# DeepHoyer: Learning Sparser Neural Network with Differentiable Scale-Invariant Sparsity Measures

**Huanrui Yang, Wei Wen, Hai Li**
Department of Electrical and Computer Engineering, Duke University, Durham, NC 27708
{huanrui.yang, wei.wen, hai.li}@duke.edu

## Abstract

In seeking for sparse and efficient neural network models, many previous works investigated on enforcing $\ell_1$ or $\ell_0$ regularizers to encourage weight sparsity during training. The $\ell_0$ regularizer measures the parameter sparsity directly and is invariant to the scaling of parameter values. But it cannot provide useful gradients and therefore requires complex optimization techniques. The $\ell_1$ regularizer is almost everywhere differentiable and can be easily optimized with gradient descent. Yet it is not scale-invariant and causes the same shrinking rate to all parameters, which is inefficient in increasing sparsity. Inspired by the Hoyer measure (the ratio between $\ell_1$ and $\ell_2$ norms) used in traditional compressed sensing problems, we present DeepHoyer, a set of sparsity-inducing regularizers that are both differentiable almost everywhere and scale-invariant. Our experiments show that enforcing DeepHoyer regularizers can produce even sparser neural network models than previous works, under the same accuracy level. We also show that DeepHoyer can be applied to both element-wise and structural pruning. The codes are available at https://github.com/yanghr/DeepHoyer.

## 1 Introduction

The use of deep neural network (DNN) models has been expanded from handwritten digit recognition (LeCun et al., 1998) to real-world applications, such as large-scale image classification (Simonyan & Zisserman, 2014), self driving (Makantasis et al., 2015) and complex control problems (Mnih et al., 2013). However, a modern DNN model like AlexNet (Krizhevsky et al., 2012) or ResNet (He et al., 2016) often introduces a large number of parameters and computation load, which makes the deployment and real-time processing on embedded and edge devices extremely difficult (Han et al., 2015b;a; Wen et al., 2016). Thus, model compression techniques, especially pruning methods that increase the sparsity of weight matrices, have been extensively studied to reduce the memory consumption and computation cost of DNNs (Han et al., 2015b;a; Wen et al., 2016; Guo et al., 2016; Louizos et al., 2017b; Luo et al., 2017; Zhang et al., 2018; Liu et al., 2015).

Most of the previous works utilize some form of sparsity-inducing regularizer in searching for sparse neural networks. The $\ell_1$ regularizer, originally proposed by Tibshirani (1996), can be easily optimized through gradient descent for its convex and almost everywhere differentiable property. Therefore it is widely used in DNN pruning: Liu et al. (2015) directly apply $\ell_1$ regularization to all the weights of a DNN to achieve element-wise sparsity; Wen et al. (2016; 2017) present structural sparsity via group lasso, which applies an $\ell_1$ regularization over the $\ell_2$ norms of different groups of parameters. However, it has been noted that the value of the $\ell_1$ regularizer is proportional to the scaling of parameters (i.e. $||\alpha W||_1 = |\alpha| \cdot ||W||_1$), so it "scales down" all the elements in the weight matrices with the same speed. This is not efficient in finding sparsity and may sacrifice the flexibility of the trained model. On the other hand, the $\ell_0$ regularizer directly reflects the real sparsity of weights and is scale invariant (i.e. $||\alpha W||_0 = ||W||_0, \forall \alpha \neq 0$), yet the $\ell_0$ norm cannot provide useful gradients. Han et al. (2015b) enforce an element-wise $\ell_0$ constraint by iterative pruning a fixed percentage of smallest weight elements, which is a heuristic method and therefore can hardly achieve optimal compression rate. Some recent works mitigate the lack of gradient information by integrating $\ell_0$ regularization with stochastic approximation (Louizos et al., 2017b) or more complex optimization methods (e.g.

ADMM) (Zhang et al., 2018). These additional measures brought overheads to the optimization process, making the use of these methods on larger networks difficult. To achieve even sparser neural networks, we argue to move beyond $\ell_0$ and $\ell_1$ regularizers and seek for a sparsity-inducing regularizer that is both almost everywhere differentiable (like $\ell_1$) and scale-invariant (like $\ell_0$).

Beyond the $\ell_1$ regularizer, plenty of non-convex sparsity measurements have been used in the field of feature selection and compressed sensing (Hurley & Rickard, 2009; Wen et al., 2018). Some popular regularizers like SCAD (Fan & Li, 2001), MDP (Zhang et al., 2010) and Trimmed $\ell_1$ (Yun et al., 2019) use a piece-wise formulation to mitigate the proportional scaling problem of $\ell_1$. The piece-wise formulation protects larger elements by having zero penalty to elements greater than a predefined threshold. However, it is extremely costly to manually seek for the optimal trimming threshold, so it is hard to obtain optimal result in DNN pruning by using these regularizers. The transformed $\ell_1$ regularizer formulated as $\sum_{i=1}^{N} \frac{(a+1)|w_i|}{a+|w_i|}$ manages to smoothly interpolate between $\ell_1$ and $\ell_0$ by tuning the hyperparameter $a$ (Ma et al., 2019). However, such an approximation is close to $\ell_0$ only when $a$ approaches infinity, so the practical formulation of the transformed $\ell_1$ (i.e. $a = 1$) is still not scale-invariant.

Particularly, we are interested in the Hoyer regularizer (Hoyer, 2004), which estimates the sparsity of a vector with the ratio between its $\ell_1$ and $\ell_2$ norms. Comparing to other sparsity-inducing regularizers, Hoyer regularizer achieves superior performance in the fields of non-negative matrix factorization (Hoyer, 2004), sparse reconstruction (Esser et al., 2013; Tran et al., 2018) and blend deconvolution (Krishnan et al., 2011; Repetti et al., 2015). We note that Hoyer regularizer is both almost everywhere differentiable and scale invariant, satisfying the desired property of a sparsity-inducing regularizer. We therefore propose *DeepHoyer*, which is the first Hoyer-inspired regularizers for DNN sparsification. Specifically, the contributions of this work include:

- *Hoyer-Square (HS) regularizer for element-wise sparsity:* We enhance the original Hoyer regularizer to the HS regularizer and achieve element-wise sparsity by applying it in the training of DNNs. The HS regularizer is both almost everywhere differentiable and scale invariant. It has the same range and minima structure as the $\ell_0$ norm. Thus, the HS regularizer presents the ability of turning small weights to zero while protecting and maintaining those weights that are larger than an induced, gradually adaptive threshold;

- *Group-HS regularizer for structural sparsity*, which is extended from the HS regularizer;

- *Generating sparser DNN models:* Our experiments show that the proposed regularizers beat state-of-the-arts in both element-wise and structural weight pruning of modern DNNs.

## 2 RELATED WORK ON DNN PRUNING

It is well known that high redundancy pervasively exists in DNNs. Consequently, pruning methods have been extensively investigated to identify and remove unimportant weights. Some heuristic pruning methods (Han et al., 2015b; Guo et al., 2016) simply remove weights in small values to generate sparse models. These methods usually require long training time without ensuring the optimality, due to the lack of theoretical understanding and well-formulated optimization (Zhang et al., 2018). Some works formulate the problem as a sparsity-inducing optimization problem, such as $\ell_1$ regularization (Liu et al., 2015; Park et al., 2016) that can be optimized using standard gradient-based algorithms, or $\ell_0$ regularization (Louizos et al., 2017b; Zhang et al., 2018) which requires stochastic approximation or special optimization techniques. We propose DeepHoyer regularizers in this work, which belong to the line of sparsity-inducing optimization research. More specific, the proposed Hoyer-Square regularizer for element-wise pruning is scale-invariant and can serve as an differentiable approximation to the $\ell_0$ norm. Furthermore, it can be optimized by gradient-based optimization methods in the same way as the $\ell_1$ regularization. With these properties, the Hoyer-Square regularizer achieves a further 38% and 63% sparsity improvement on LeNet-300-100 model and LeNet-5 model respectively comparing to previous state-of-the-arts, and achieves the highest sparsity on AlexNet without accuracy loss.

Structurally sparse DNNs attempt to create regular sparse patterns that are friendly for hardware execution. To achieve the goal, Li et al. (2016) propose to remove filters with small norms; Wen et al. (2016) apply group Lasso regularization based methods to remove various structures (e.g., filters, channels, layers) in DNNs and the similar approaches are used to remove neurons (Alvarez &

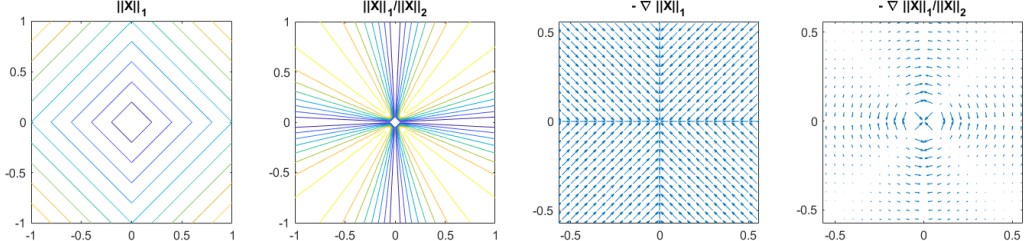

Figure 1: Comparing the $\ell_1$ and the Hoyer regularizer of a 2-D vector. Their contours are shown in the left 2 subplots (darker color corresponds to a lower value). The right 2 subplots compare their negative gradients.

Salzmann, 2016); Liu et al. (2017) and Gordon et al. (2018) (MorphNet) enforce sparsity-inducing regularization on the scaling parameters within Batch Normalization layers to remove the corresponding channels in DNNs; ThiNet (Luo et al., 2017) removes unimportant filters by minimizing the reconstruction error of feature maps; and He et al. (2017) incorporate both Lasso regression and reconstruction error into the optimization problem. Bayesian optimization methods have also been applied for neuron pruning (Louizos et al., 2017a; Neklyudov et al., 2017), yet these methods are not applicable in large-scale problems like ImageNet. We further advance the DeepHoyer to learn structured sparsity (such as reducing filters and channels) with the newly proposed "Group-HS" regularization. The Group-HS regularizer further improves the computation reduction of the LeNet-5 model by 8.8% from the $\ell_1$ based method (Wen et al., 2016), and by 110.6% from the $\ell_0$ based method (Louizos et al., 2017b). Experiments on ResNet models reveal that the accuracy-speedup tradeoff induced by Group-HS constantly stays above the Pareto frontier of previous methods. More detailed results can be found in Section 5.

## 3 MEASURING SPARSITY WITH THE HOYER MEASURE

Sparsity measures provide tractable sparsity constraints for enforcement during problem solving and therefore have been extensively studied in the compressed sensing society. In early non-negative matrix factorization (NMF) research, a consensus was that a sparsity measure should map a $n$-dimensional vector $X$ to a real number $S \in [0, 1]$, such that the possible sparsest vectors with only one nonzero element has $S = 1$, and a vector with all equal elements has $S = 0$ (Hoyer, 2004). Unders the assumption, the Hoyer measure was proposed as follows

$$S(X) = \frac{\sqrt{n} - (\sum_i |x_i|)/\sqrt{\sum_i x_i^2}}{\sqrt{n} - 1}. \tag{1}$$

It can be seen that

$$1 \leq \frac{\sum_i |x_i|}{\sqrt{\sum_i x_i^2}} \leq \sqrt{n}, \quad \forall X \in \mathbb{R}^n. \tag{2}$$

Thus, the normalization in Equation (1) fits the measure $S(X)$ into the $[0, 1]$ interval. According to the survey by Hurley & Rickard (2009), among the six desired heuristic criteria of sparsity measures, the Hoyer measure satisfies five, more than all other commonly applied sparsity measures. Given its success as a sparsity measure in NMF, the Hoyer measure has been applied as a sparsity-inducing regularizer in optimization problems such as blind deconvolution (Repetti et al., 2015) and image deblurring (Krishnan et al., 2011). Without the range constraint, the Hoyer regularizer in these works adopts the form $R(X) = \frac{\sum_i |x_i|}{\sqrt{\sum_i x_i^2}}$ directly, as the ratio of the $\ell_1$ and $\ell_2$ norms.

Figure 1 compares the Hoyer regularizer and the $\ell_1$ regularizer. Unlike the the $\ell_1$ norm with a single minimum at the origin, the Hoyer regularizer has minima along axes, the structure of which is very similar to the $\ell_0$ norm's. Moreover, the Hoyer regularizer is scale-invariant, i.e. $R(\alpha X) = R(X)$, because both the $\ell_1$ norm and the $\ell_2$ norm are proportional to the scale of $X$. The gradients of the Hoyer regularizer are purely radial, leading to "rotations" towards the nearest axis. These features make the Hoyer regularizer outperform the $\ell_1$ regularizer on various tasks (Esser et al., 2013; Tran

et al., 2018; Krishnan et al., 2011; Repetti et al., 2015). The theoretical analysis by Yin et al. (2014) also proves that the Hoyer regularizer has a better guarantee than the $\ell_1$ norm on recovering sparse solutions from coherent and redundant representations.

# 4 MODEL COMPRESSION WITH DEEPHOYER REGULARIZERS

Inspired by the Hoyer regularizer, we propose two types of DeepHoyer regularizers: the *Hoyer-Square* regularizer (HS) for element-wise pruning and the *Group-HS* regularizer for structural pruning.

## 4.1 HOYER-SQUARE REGULARIZER FOR ELEMENT-WISE PRUNING

Since the process of the element-wise pruning is equivalent to regularizing each layer's weight with the $\ell_0$ norm, it is intuitive to configure the sparsity-inducing regularizer to have a similar behavior as the $\ell_0$ norm. As shown in Inequality (2), the value of the original Hoyer regularizer of a $N$-dimensional nonzero vector lies between 1 and $\sqrt{N}$, while its $\ell_0$ norm is within the range of $[1, N]$. Thus we propose to apply the square of Hoyer regularizer, namely *Hoyer-Square* (HS), to the weights $W$ of a layer, like

$$H_S(W) = \frac{(\sum_i |w_i|)^2}{\sum_i w_i^2}. \tag{3}$$

The proposed HS regularizer behaves as a differentiable approximation to the $\ell_0$ norm. First, both regularizers now have the same range of $[1, N]$. Second, $H_S$ is scale invariant as $H_S(\alpha W) = H_S(W)$ holds for $\forall \alpha \neq 0$, so as the $\ell_0$ norm. Moreover, as the squaring operator monotonously increases in the range of $[1, \sqrt{N}]$, the Hoyer-Square regularizer's minima remain along the axes as the Hoyer regularizer's do (see Figure 1). In other words, they have similar minima structure as the $\ell_0$ norm. At last, the Hoyer-Square regularizer is also almost everywhere differentiable and Equation (4) formulates the gradient of $H_S$ w.r.t. an element $w_j$ in the weight matrix $W$:

$$\partial_{w_j} H_S(W) = 2 sign(w_j) \frac{\sum_i |w_i|}{(\sum_i w_i^2)^2} (\sum_i w_i^2 - |w_j| \sum_i |w_i|). \tag{4}$$

Very importantly, this formulation induces a trimming effect: when $H_S(W)$ is being minimized through gradient descent, $w_j$ moves towards 0 if $|w_j| < \frac{\sum_i w_i^2}{\sum_i |w_i|}$, otherwise moves away from 0. In other words, unlike the $\ell_1$ regularizer which tends to shrink all elements, our Hoyer-Square regularizer will turn weights in small value to zero meanwhile protecting large weights. Traditional trimmed regularizers (Fan & Li, 2001; Zhang et al., 2010; Yun et al., 2019) usually define a trimming threshold as a fixed value or percentage. Instead, the HS regularizer can gradually extend the scope of pruning as more weights coming close to zero. This behavior can be observed in the gradient descent path shown in Figure 2.

## 4.2 GROUP-HS REGULARIZER FOR STRUCTURAL PRUNING

Beyond element-wise pruning, structural pruning is often more preferred because it can construct the sparsity in a structured way and therefore achieve higher computation speed-up on general computation platforms (Wen et al., 2016). The structural pruning is previously empowered by the *group lasso* (Yuan & Lin, 2006; Wen et al., 2016), which is the sum (i.e. $\ell_1$ norm) of the $\ell_2$ norms of all the groups within a weight matrix like

$$R_G(W) = \sum_{g=1}^{G} ||w^{(g)}||_2, \tag{5}$$

where $||W||_2 = \sqrt{\sum_i w_i^2}$ represents the $\ell_2$ norm, $w^{(g)}$ is a group of elements in the weight matrix $W$ which consists of $G$ such groups.

Following the same approach in Section 4.1, we use the Hoyer-Square regularizer to replace the $\ell_1$ regularizer in the group lasso formulation and define the *Group-HS* ($G_H$) regularizer in Equation (6):

$$G_H(W) = \frac{(\sum_{g=1}^{G} ||w^{(g)}||_2)^2}{\sum_{g=1}^{G} ||w^{(g)}||_2^2} = \frac{(\sum_{g=1}^{G} ||w^{(g)}||_2)^2}{||W||_2^2}. \tag{6}$$

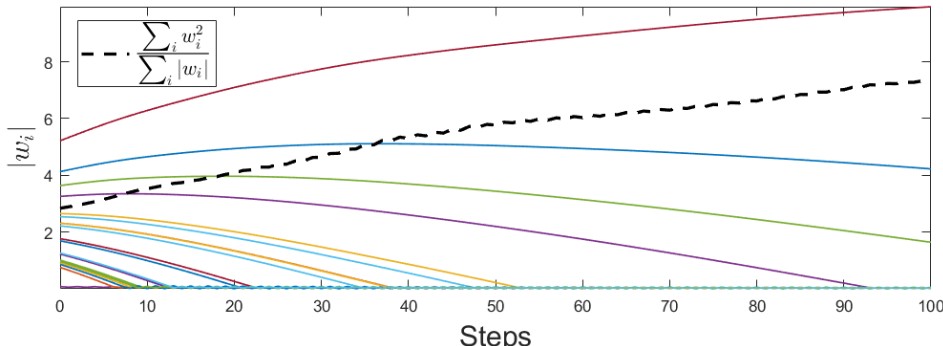

Figure 2: Minimization path of Hoyer-Square regularizer during gradient descent, with $W \in \mathbb{R}^{20}$ initialized as i.i.d. $\mathcal{N}(0, 1)$. The figure shows the path of each element $w_i$ during the minimization, with the black dash line showing the induced trimming threshold.

Note that the second equality holds when and only when the groups cover all the elements of $W$ without overlapping with each other. Our experiments in this paper satisfy this requirement. However, the Group-HS regularizer can always be used in the form of the first equality when overlapping exists across groups. The gradient and the descent path of the Group-HS regularizer are very similar to those of the Hoyer-Square regularizer, and therefore we omit the detailed discussion here. The derivation of the Group-HS regularizer's gradient shall be found in **Appendix A**.

### 4.3 APPLY DEEPHOYER REGULARIZERS IN DNN TRAINING

The deployment of the DeepHoyer regularizers in DNN training follows the common layer-based regularization approach (Wen et al., 2016; Liu et al., 2015). For element-wise pruning, we apply the Hoyer-Square regularizer to layer weight matrix $W^{(l)}$ for all $L$ layers, and directly minimize it alongside the DNN's original training objective $\mathcal{L}(W^{(1:L)})$. The $\ell_2$ regularizer can also be added to the objective if needed. Equation (7) presents the training objective with $H_S$ defined in Equation (3). Here, $\alpha$ and $\beta$ are pre-selected weight decay parameters for the regularizers.

$$\min_{W^{(1:L)}} \mathcal{L}(W^{(1:L)}) + \sum_{l=1}^{L} (\alpha H_S(W^{(l)}) + \beta ||W^{(l)}||_2). \tag{7}$$

For structural pruning, we mainly focus on pruning the columns and rows of fully connected layers and the filters and channels of convolutional layers. More specific, we group a layer in filter-wise and channel-wise fashion as proposed by Wen et al. (2016) and then apply the Group-HS regularizer to the layer. The resulted optimization objective is formulated in Equation (8).

$$\min_{W^{(1:L)}} \mathcal{L}(W^{(1:L)}) + \sum_{l=1}^{L} (\alpha_n \frac{(\sum_{n_l=1}^{N_l} ||w_{n_l,:,:,:}^{(l)}||_2)^2}{||W^{(l)}||_2^2} + \alpha_c \frac{(\sum_{c_l=1}^{C_l} ||w_{:,c_l,:,:}^{(l)}||_2)^2}{||W^{(l)}||_2^2} + \beta ||W^{(l)}||_2). \tag{8}$$

Here $N_l$ is the number of filters and $C_l$ is the number of channels in the $l^{th}$ layer if it is a convolutional layer. If the $l^{th}$ layer is fully connected, then $N_l$ and $C_l$ is the number of rows and columns respectively. $\alpha_n$, $\alpha_c$ and $\beta$ are pre-selected weight decay parameters for the regularizers.

The recent advance in stochastic gradient descent (SGD) method provides satisfying results under large-scale non-convex settings (Sutskever et al., 2013; Kingma & Ba, 2014), including DNNs with non-convex objectives (Auer et al., 1996). So we can directly optimize the DeepHoyer regularizers with the same SGD optimizer used for the original DNN training objective, despite their nonconvex formulations. Our experiments show that the tiny-bit nonconvexity induced by DeepHoyer does not affect the performance of DNNs.

The pruning is conducted by following the common three-stage operations: (1) train the DNN with the DeepHoyer regularizer, (2) prune all the weight elements smaller than a predefined small threshold, and (3) finetune the model by fixing all the zero elements and removing the DeepHoyer regularizer.

Table 1: Element-wise pruning results on LeNet-300-100 model @ accuracy 98.4%

| Method | Nonzero wights left after pruning | | | |
|---|---|---|---|---|
| | Total | FC1 | FC2 | FC3 |
| Orig | 266.2k | 235.2k | 30k | 1k |
| (Han et al., 2015b) | 21.8k (8%) | 18.8k (8%) | 2.7k (9%) | 260 (26%) |
| (Zhang et al., 2018) | 11.6k (4.37%) | 9.4k (4%) | 2.1k (7%) | 120 (12%) |
| (Lee et al., 2019) | 13.3k (5.0%) | Not reported in (Lee et al., 2019) | | |
| (Ma et al., 2019)[1] | 6.4k (2.40%) | 5.0k (2.11%) | 1.2k (4.09%) | 209 (20.90%) |
| Hoyer | 6.0k (2.27%) | 5.3k (2.25%) | **672 (2.24%)** | **82 (8.20%)** |
| Hoyer-Square | **4.6k (1.74%)** | **3.7k (1.57%)** | 768 (2.56%) | 159 (15.90%) |

Table 2: Element-wise pruning results on LeNet-5 model @ accuracy 99.2%

| Method | Nonzero wights left after pruning | | | | |
|---|---|---|---|---|---|
| | Total | CONV1 | CONV2 | FC1 | FC2 |
| Orig | 430.5k | 500 | 25k | 400k | 5k |
| (Han et al., 2015b) | 36k (8%) | 330 (66%) | 3k (12%) | 32k (8%) | 950 (19%) |
| (Zhang et al., 2018) | 6.1k (1.4%) | 100 (20%) | 2k (8%) | 3.6k (0.9%) | 350 (7%) |
| (Lee et al., 2019) | 8.6k (2.0%) | Not reported in (Lee et al., 2019) | | | |
| (Ma et al., 2019)[1] | 5.4k (1.3%) | 100 (20%) | 690 (2.8%) | 4.4k (1.1%) | 203 (4.1%) |
| Hoyer | 4.0k (0.9%) | **53 (10.6%)** | **613 (2.5%)** | 3.2k (0.8%) | **136 (2.7%)** |
| Hoyer-Square | **3.5k (0.8%)** | 67 (13.4%) | 848 (3.4%) | **2.4k (0.6%)** | 234 (4.7%) |

## 5 EXPERIMENT RESULT

The proposed DeepHoyer regularizers are first tested on the MNIST benchmark using the LeNet-300-100 fully connected model and the LeNet-5 CNN model (LeCun et al., 1998). We also conduct tests on the CIFAR-10 dataset (Krizhevsky & Hinton, 2009) with ResNet models (He et al., 2016) in various depths, and on ImageNet ILSVRC-2012 benchmark (Russakovsky et al., 2015) with the AlexNet model (Krizhevsky et al., 2012) and the ResNet-50 model (He et al., 2016). All the models are implemented and trained in the PyTorch deep learning framework (Paszke et al., 2017), where we match the model structure and the benchmark performance with those of previous works for the fairness of comparison. The experiment results presented in the rest of this section show that the proposed DeepHoyer regularizers consistently outperform previous works in both element-wise and structural pruning. Detailed information on the experiment setups and the parameter choices of our reported results can be found in **Appendix B**.

### 5.1 ELEMENT-WISE PRUNING

Table 1 and Table 2 summarize the performance of the proposed Hoyer-square regularizer on the MNIST benchmark, with comparisons against state of the art (SOTA) element-wise pruning methods. Without losing the testing accuracy, training with the Hoyer-Square regularizer reduces the number of nonzero weights by $54.5\times$ on the LeNet-300-100 model and by $122\times$ on the LeNet-5 model. Among all the methods, ours achieves the highest sparsity: it is a $38\%$ improvement on the LeNet-300-100 model and a $63\%$ improvement on the LeNet-5 model comparing to the best available methods. Additional results in **Appendix C.1** further illustrates the effect of the Hoyer-Square regularizer on each layer's weight distribution during the training process.

The element-wise pruning performance on the AlexNet model testing on the ImageNet benchmark is presented in Table 3. Without losing the testing accuracy, the Hoyer-Square regularizer improves

---

[1]We implement the transformed $\ell_1$ regularizer in (Ma et al., 2019) ourselves because the experiments in the original paper are under different settings. Implementation details can be found in **Appendix B**.

Table 3: Element-wise pruning results on AlexNet model.

| Method | Top-5 error increase | #Parameters | Percentage left |
|---|---|---|---|
| Orig | +0.0% | 60.9M | 100% |
| (Han et al., 2015b) | -0.1% | 6.7M | 11.0% |
| (Guo et al., 2016) | +0.2% | 3.45M | 5.67% |
| (Dai et al., 2017) | -0.1% | 3.1M | 6.40% |
| (Ma et al., 2019)[1] | +0.0% | 3.05M | 5.01% |
| (Zhang et al., 2018) | +0.0% | 2.9M | 4.76% |
| Hoyer | +0.0% | 3.62M | 5.94% |
| Hoyer-Square | +0.0% | **2.85M** | **4.69**% |

Table 4: Structural pruning results on LeNet-300-100 model

| Method | Accuracy | #FLOPs | Pruned structure |
|---|---|---|---|
| Orig | 98.4% | 266.2k | 784-300-100 |
| Sparse VD (Molchanov et al., 2017) | 98.2% | 67.3k (25.28%) | 512-114-72 |
| BC-GNJ (Louizos et al., 2017a) | 98.2% | 28.6k (10.76%) | 278-98-13 |
| BC-GHS (Louizos et al., 2017a) | 98.2% | 28.1k (10.55%) | 311-86-14 |
| $\ell_{0_{hc}}$ (Louizos et al., 2017b) | 98.2% | 26.6k (10.01%) | 266-88-33 |
| Bayes $\ell_{1_{trim}}$ (Yun et al., 2019) | 98.3% | 20.5k (7.70%) | 245-75-25 |
| Group-HS | 98.2% | **16.5k (6.19%)** | 353-45-11 |

the compression rate by $21.3\times$. This result is the highest among all methods, even better than the ADMM method (Zhang et al., 2018) which requires two additional Lagrange multipliers and involves the optimization of two objectives. Considering that the optimization of the Hoyer-Square regularizer can be directly realized on a single objective without additional variables, we conclude that the Hoyer-Square regularizer can achieve a sparse DNN model with a much lower cost. A more detailed layer-by-layer sparsity comparison of the compressed model can be found in **Appendix C.2**.

We perform the ablation study for performance comparison between the Hoyer-Square regularizer and the original Hoyer regularizer. The results in Tables 1, 2 and 3 all show that the Hoyer-Square regularizer always achieves a higher compression rate than the original Hoyer regularizer. The layer-wise compression results show that the Hoyer-Square regularizer emphasizes more on the layers with more parameters (i.e. FC1 for the MNIST models). This corresponds to the fact that the value of the Hoyer-Square regularizer is proportional to the number of non-zero elements in the weight. These observations validate our choice to use the Hoyer-Square regularizer for DNN compression.

## 5.2 STRUCTURAL PRUNING

This section reports the effectiveness of the Group-HS regularizer in structural pruning tasks. Here we mainly focus on the number of remaining neurons (output channels for convolution layers and rows for fully connected layers) after removing the all-zero channels or rows in the weight matrices. The comparison is then made based on the required float-point operations (FLOPs) to inference with the remaining neurons, which indeed represents the potential inference speed of the pruned model. As shown in Table 4, training with the Group-HS regularizer can reduce the number of FLOPs by $16.2\times$ for the LeNet-300-100 model with a slight accuracy drop. This is the highest speedup among all existing methods achieving the same testing accuracy. Table 5 shows that the Group-HS regularizer can reduce the number of FLOPs of the LeNet-5 model by $12.4\times$, which outperforms most of the existing work—an 8.8% increase from the $\ell_1$ based method (Wen et al., 2016) and a 110.6% increase from the $\ell_0$ based method (Louizos et al., 2017b). Only the Bayesian compression (BC) method with the group-horseshoe prior (BC-GHS) (Louizos et al., 2017a) achieves a slightly higher speedup on the LeNet-5 model. However, the complexity of high dimensional Bayesian inference limits BC's capability. It is difficult to apply BC to ImageNet-level problems and large DNN models like ResNet.

Table 5: Structural pruning result on LeNet-5 model

| Method | Accuracy | #FLOPs | Pruned structure |
|---|---|---|---|
| Orig | 99.2% | 2293k | 20-50-800-500 |
| Sparse VD (Molchanov et al., 2017) | 99.0% | 660.2k (28.79%) | 14-19-242-131 |
| GL (Wen et al., 2016) | 99.0% | 201.8k (8.80%) | 3-12-192-500 |
| SBP (Neklyudov et al., 2017) | 99.1% | 212.8k (9.28%) | 3-18-284-283 |
| BC-GNJ (Louizos et al., 2017a) | 99.0% | 282.9k (12.34%) | 8-13-88-13 |
| BC-GHS (Louizos et al., 2017a) | 99.0% | **153.4k (6.69%)** | 5-10-76-16 |
| $\ell_{0_{hc}}$ (Louizos et al., 2017b) | 99.0% | 390.7k (17.04%) | 9-18-26-25 |
| Bayes $\ell_{1_{trim}}$ (Yun et al., 2019) | 99.0% | 334.0k (14.57%) | 8-17-53-19 |
| Group-HS | 99.0% | **169.9k (7.41%)** | 5-12-139-13 |

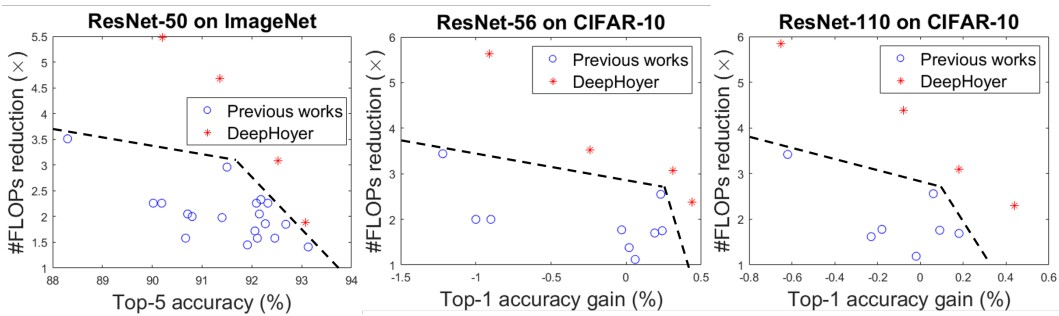

Figure 3: Comparisons of accuracy-#FLOPs tradeoff on ImageNet and CIFAR-10, black dash lines mark the Pareto frontiers. The exact data for the points are listed in **Appendix C.3**.

In contrast, the effectiveness of the Group-HS regularizer can be easily extended to deeper models and larger datasets, which is demonstrated by our experiments. We apply the Group-HS regularizer to ResNet models (He et al., 2016) on the CIFAR-10 and the ImageNet datasets. Pruning ResNet has long been considered difficult due to the compact structure of the ResNet model. Since previous works usually report the compression rate at different accuracy, we use the "accuracy-#FLOPs" plot to represent the tradeoff. The tradeoff between the accuracy and the FLOPs are explored in this work by changing the strength of the Group-HS regularizer used in training. Figure 3 shows the performance of DeepHoyer constantly stays above the Pareto frontier of previous methods.

## 6 CONCLUSIONS

In this work, we propose *DeepHoyer*, a set of sparsity-inducing regularizers that are both scale-invariant and almost everywhere differentiable. We show that the proposed regularizers have similar range and minima structure as the $\ell_0$ norm, so it can effectively measure and regularize the sparsity of the weight matrices of DNN models. Meanwhile, the differentiable property enables the proposed regularizers to be simply optimized with standard gradient-based methods, in the same way as the $\ell_1$ regularizer is. In the element-wise pruning experiment, the proposed Hoyer-Square regularizer achieves a $38\%$ sparsity increase on the LeNet-300-100 model and a $63\%$ sparsity increase on the LeNet-5 model without accuracy loss comparing to the state-of-the-art. A $21.3\times$ model compression rate is achieved on AlexNet, which also surpass all previous methods. In the structural pruning experiment, the proposed Group-HS regularizer further reduces the computation load by 24.4% from the state-of-the-art on LeNet-300-100 model. It also achieves a 8.8% increase from the $\ell_1$ based method and a 110.6% increase from the $\ell_0$ based method of the computation reduction rate on the LeNet-5 model. For CIFAR-10 and ImageNet dataset, the accuracy-FLOPs tradeoff achieved by training ResNet models with various strengths of the Group-HS regularizer constantly stays above the Pareto frontier of previous methods. These results prove that the DeepHoyer regularizers are effective in achieving both element-wise and structural sparsity in deep neural networks, and can produce even sparser DNN models than previous works.

ACKNOWLEDGMENTS

The authors would like to thank Feng Yan for his help on computation resources throughout this project. Our work was supported in part by NSF SPX-1725456 and NSF CNS-1822085.

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

## A    DERIVATION OF DEEPHOYER REGULARIZERS' GRADIENTS

In this section we provide detailed derivation of the gradient of the Hoyer-Square regularizer and the Group-GS regularizer w.r.t. an element $w_j$ in the weight matrix $W$.

The gradient of the Hoyer-Square regularizer is shown in Equation (9). The formulation shown in Equation (4) is achieved at the end of the derivation.

$$
\begin{aligned}
\partial_{w_j} H_S(W) &= \frac{[\partial_{w_j}((\sum_i |w_i|)^2)]\sum_i w_i^2 - [\partial_{w_j}(\sum_i w_i^2)](\sum_i |w_i|)^2}{(\sum_i w_i^2)^2} \\
&= \frac{2[\partial_{w_j}(|w_j|)]\sum_i |w_i|\sum_i w_i^2 - 2w_j(\sum_i |w_i|)^2}{(\sum_i w_i^2)^2} \\
&= 2\frac{\sum_i |w_i|}{(\sum_i w_i^2)^2}(sign(w_j)\sum_i w_i^2 - sign(w_j)|w_j|\sum_i |w_i|) \\
&= 2sign(w_j)\frac{\sum_i |w_i|}{(\sum_i w_i^2)^2}(\sum_i w_i^2 - |w_j|\sum_i |w_i|).
\end{aligned}
\tag{9}
$$

The gradient of the Group-HS regularizer is shown in Equation (10). For simplicity we use the form shown in the second equality of Equation (6), where there is no overlapping between the groups. Here we assume that $w_j$ belongs to group $w^{(\hat{g})}$.

$$
\begin{aligned}
\partial_{w_j} G_H(W) &= \partial_{w_j}\frac{(\sum_{g=1}^G ||w^{(g)}||_2)^2}{\sum_i w_i^2} \\
&= \frac{[\partial_{w_j}((\sum_{g=1}^G ||w^{(g)}||_2)^2)]\sum_i w_i^2 - [\partial_{w_j}(\sum_i w_i^2)](\sum_{g=1}^G ||w^{(g)}||_2)^2}{(\sum_i w_i^2)^2} \\
&= \frac{2[\partial_{w_j}(||w^{(\hat{g})}||_2)]\sum_{g=1}^G ||w^{(g)}||_2\sum_i w_i^2 - 2w_j(\sum_{g=1}^G ||w^{(g)}||_2)^2}{(\sum_i w_i^2)^2} \\
&= 2\frac{\sum_{g=1}^G ||w^{(g)}||_2}{(\sum_i w_i^2)^2}(\frac{w_j}{||w^{(\hat{g})}||_2}\sum_i w_i^2 - w_j\sum_{g=1}^G ||w^{(g)}||_2) \\
&= 2\frac{w_j}{||w^{(\hat{g})}||_2}\frac{\sum_i |w_i|}{(\sum_i w_i^2)^2}(\sum_i w_i^2 - ||w^{(\hat{g})}||_2\sum_{g=1}^G ||w^{(g)}||_2).
\end{aligned}
\tag{10}
$$

## B    DETAILED EXPERIMENT SETUP

### B.1    MNIST EXPERIMENTS

The MNIST dataset (LeCun et al., 1998) is a well known handwritten digit dataset consists of grey-scale images with the size of $28 \times 28$ pixels. We use the dataset API provided in the "torchvision" python package to access the dataset. In our experiments we use the whole 60,000 training set images for the training and the whole 10,000 testing set images for the evaluation. All the accuracy results reported in the paper are evaluated on the testing set. Both the training set and the testing set are normalized to have zero mean and variance one. Adam optimizer (Kingma & Ba, 2014) with learning rate 0.001 is used throughout the training process. All the MNIST experiments are done with a single TITAN XP GPU.

Both the LeNet-300-100 model and the LeNet-5 model are firstly pretrained without the sparsity-inducing regularizer, where they achieve the testing accuracy of 98.4% and 99.2% respectively. Then the models are further trained for 250 epochs with the DeepHoyer regularizers applied in the objective. The weight decay parameters ($\alpha$s in Equation (7) and (8)) are picked by hand to reach the best result. In the last step, we prune the weight of each layer with threshold proportional to the standard derivation of each layer's weight. The threshold/std ratio is chosen to achieve the highest sparsity without accuracy loss. All weight elements with a absolute value smaller than the threshold

Table 6: Hyper parameter used for MNIST benchmarks

| Model | LeNet-300-100 | | LeNet-5 | |
|---|---|---|---|---|
| Regularizer | Decay | Threshold/std | Decay | Threshold/std |
| Hoyer | 0.02 | 0.05 | 0.01 | 0.08 |
| Hoyer-Square | 0.0002 | 0.03 | 0.0001 | 0.03 |
| Group-HS | 0.002 | 0.8 | 0.1 | 0.008 |
| Transformed $\ell_1$ | 2e-5 | 0.3 | 2e-5 | 0.6 |

is set to zero and is fixed during the final finetuning. The pruned model is finetuned for another 100 steps without DeepHoyer regularizers and the best testing accuracy achieved is reported. Detailed parameter choices used in achieving the reported results are listed in Table 6.

## B.2 IMAGENET AND CIFAR-10 EXPERIMENTS

The ImageNet dataset is a large-scale color-image dataset containing 1.2 million images of 1000 categories (Russakovsky et al., 2015), which has long been utilized as an important benchmark on image classification problems. In this paper, we use the "ILSVRC2012" version of the dataset, which can be found at http://www.image-net.org/challenges/LSVRC/2012/nonpub-downloads. We use all the data in the provided training set to train our model, and use the provided validation set to evaluate our model and report the testing accuracy. We follow the data reading and preprocessing pipeline suggested by the official PyTorch ImageNet example (https://github.com/pytorch/examples/tree/master/imagenet). For training images, we first randomly crop the training images to desired input size, then apply random horizontal flipping and finally normalize them before feeding them into the network. Validation images are first resized to $256 \times 256$ pixels, then center cropped to desired input size and normalized in the end. We use input size $227 \times 227$ pixels for experiments on the AlexNet, and input size $224 \times 224$ for experiments on the ResNet-50. All the models are optimized with the SGD optimizer Sutskever et al. (2013), and the batch size is chosen as 256 for all the experiments. Two TITAN XP GPUs are used in parallel for the AlexNet training and four are used for the ResNet-50 training.

One thing worth noticing is that the AlexNet model provided in the "torchvision" package is not the ordinary version used in previous works Han et al. (2015b); Wen et al. (2016); Zhang et al. (2018). Therefore we reimplement the AlexNet model in PyTorch for fair comparison. We pretrain the implemented model for 90 epochs and achieve 19.8 % top-5 error, which is the same as reported in previous works. In the AlexNet experiment, the reported result in Table 3 is achieved by applying the Hoyer-Square regularizer with decay parameter 1e-6. Before the pruning, the model is firstly train from the pretrained model with the Hoyer-Square regularizer for 90 epochs, where an initial learning rate 0.001 is used. An $\ell_2$ regularization with 1e-4 decay is also applied. We then prune the convolution layers with threshold 1e-4 and the FC layers with threshold equal to $0.4\times$ of their standard derivations. The model is then finetuned until the best accuracy is reached. The learning rate is decayed by 0.1 for every 30 epochs of training. The training process with the Hoyer regularizer and the $T\ell_1$ regularizer (Ma et al., 2019) is the same as the HS regularizer. For the reported result, we use decay 1e-3 and FC threshold $0.8\times$ std for the Hoyer regularizer, and use decay 2e-5 and FC threshold $1.0\times$ std for the $T\ell_1$ regularizer.

For the ResNet-50 experiments on ImageNet, the model architecture and pretrained model provided in the "torchvision" package is directly utilized, which achieves 23.85% top-1 error and 7.13% top-5 error. All the reported results in Figure 3 and Table 8 are achieved with 90 epochs of training with the Group-HS regularizer from the pretrained model using initial learning rate 0.1. All the models are pruned with 1e-4 as threshold and finetuned to the best accuracy. We only tune the decay parameter of the Group-HS regularizer to explore the accuracy-FLOPs tradeoff. The exact decay parameter used for each result is specified in Table 8.

We also use the CIFAR-10 dataset (Krizhevsky & Hinton, 2009) to evaluate the structural pruning performance on ResNet-56 and ResNet-110 models. The CIFAR-10 dataset can be directly accessed through the dataset API provided in the "torchvision" python package. Standard preprocessing,

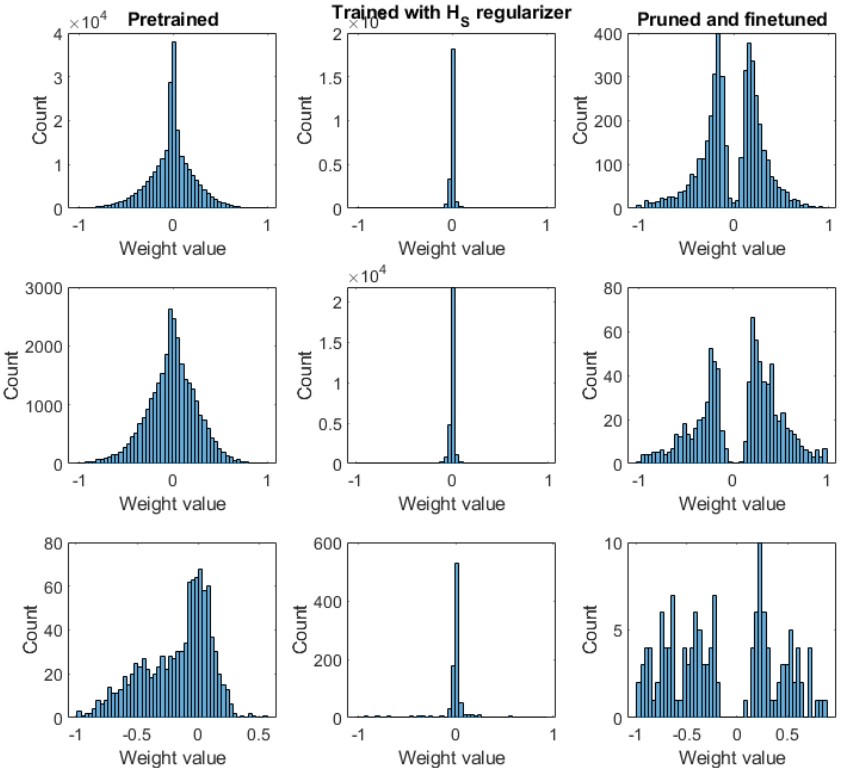

Figure 4: Histogram of *nonzero* weight elements of each layer in the LeNet-300-100 model. From top to bottom corresponds to layer FC1, FC2, FC3 respectively. The original pretrained model is shown in column 1, column 2 shows the model achieved after $H_S$ regularization, column 3 shows the final model after pruning and finetuning.

including random crop, horizontal flip and normalization is used on the training set to train the model. We implemented the ResNet models for CIFAR-10 following the description in (He et al., 2016), and pretrain the models for 164 epochs. Learning rate is set to 0.1 initially, and decayed by 0.1 at epoch 81 and epoch 122. The pretrained ResNet-56 model reaches the testing accuracy of 93.14 %, while the ResNet-110 model reaches 93.62 %. Similar to the ResNet-50 experiment, we start with the pretrained models and train with the Group-HS regularizer. Same learning rate scheduling is used for both pretraining and training with Group-HS. All the models are pruned with 1e-4 as threshold and finetuned to the best accuracy. The decay parameters of the Group-HS regularizer used to get the result in Figure 3 is specified in Table 9 and Table 10.

## C  ADDITIONAL EXPERIMENT RESULTS

### C.1  WEIGHT DISTRIBUTION AT DIFFERENT STAGES

Here we demonstrate how will the weight distribution change in each layer at different stages of our element-wise pruning process. Since most of the weight elements will be zero in the end, we only plot the histogram of *nonzero* weight elements for better observation. The histogram of each layer of the LeNet-300-100 model and the LeNet-5 model are visualized in Figure 4 and Figure 5 respectively. It can be seen that majority of the weights will be concentrated near zero after applying the $H_S$ regularizer during training, while rest of the weight elements will spread out in a wide range. The weights close to zero are then set to be exactly zero, and the model is finetuned with zero weights fixed. The resulted histogram shows that most of the weights are pruned away, only a small amount of nonzero weights are remaining in the model.

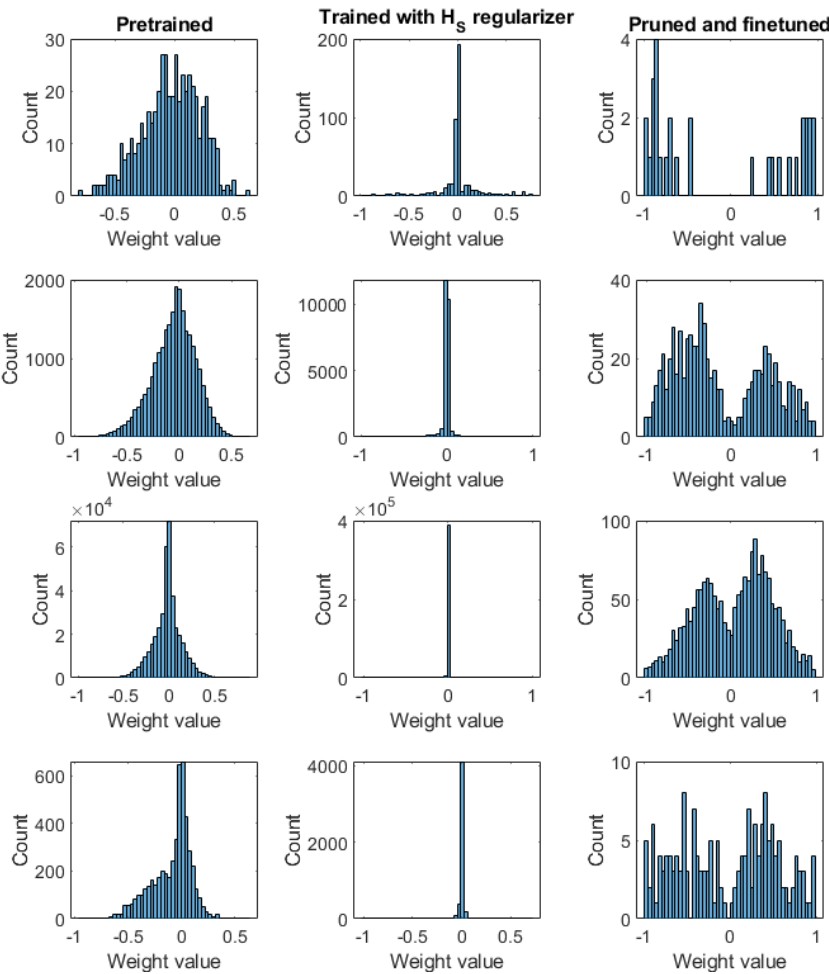

Figure 5: Histogram of *nonzero* weight elements of each layer in the LeNet-5 model. From top to bottom corresponds to layer CONV1, CONV2, FC1, FC2 respectively. The original pretrained model is shown in column 1, column 2 shows the model achieved after $H_S$ regularization, column 3 shows the final model after pruning and finetuning.

## C.2 LAYER-BY-LAYER COMPARISON OF ELEMENT-WISE PRUNING RESULT OF ALEXNET

Table 7 compares the element-wise pruning result of the Hoyer-Square regularizer on AlexNet with other methods in a layer-by-layer fashion. It can be seen that the Hoyer-Square regularizer achieves high pruning rates on the largest layers (i.e. FC1-3). This observation is consistent with the observation made on the element-wise pruning performance of models on the MNIST dataset.

## C.3 DETAILED RESULTS OF THE RESNET EXPERIMENTS

In this section we list the data used to plot Figure 3. Table 8 shows the result of pruning ResNet-50 model on ImageNet, Table 9 shows the result of pruning ResNet-56 model on CIFAR-10 and Table 10 shows the result of pruning ResNet-110 model on CIFAR-10. For all the tables, the results of previous works are listed on the top, and are ordered based on publication year. Results achieved with the Group-HS regularizer are listed below, marked with the regularization strength used for the training.

Table 7: Element-wise pruning results on AlexNet without accuracy loss. Refer to Table 3 for the full reference of the mentined methods.

| Layer | Nonzero wights left after pruning | | | | | |
|---|---|---|---|---|---|---|
| | Baseline | Han et al. | Zhang et al. | Ma et al. | Hoyer | HS |
| CONV1 | 34.8K | 29.3K | 28.2K | 24.2K | **21.3K** | 31.6K |
| CONV2 | 307.2K | 116.7K | **61.4K** | 109.9K | 77.2K | 148.4K |
| CONV3 | 884.7K | 309.7K | **168.1K** | 241.2K | 192.0K | 299.3K |
| CONV4 | 663.5K | 245.5K | **132.7K** | 207.4K | 182.6K | 275.6K |
| CONV5 | 442.2K | 163.7K | **88.5K** | 134.7K | 116.6K | 197.1K |
| FC1 | 37.7M | 3.40M | 1.06M | **0.763M** | 1.566M | **0.781M** |
| FC2 | 16.8M | 1.51M | 0.99M | 1.070M | 0.974M | **0.650M** |
| FC3 | 4.10M | 1.02M | 0.38M | 0.505M | 0.490M | **0.472M** |
| Total | 60.9M | 6.8M | 2.9M | 3.05M | 3.62M | **2.85M** |

Table 8: Structural pruning result of the ResNet-50 model on imageNet.

| Model | Top-1 acc | Top-5 acc | #FLOPs reduction |
|---|---|---|---|
| Orig | 76.15% | 92.87% | 1.00× |
| Channel pruning (He et al., 2017) | N/A | 90.80% | 2.00× |
| ThiNet-70 (Luo et al., 2017) | 72.04% | 90.67% | 1.58× |
| ThiNet-50 (Luo et al., 2017) | 71.01% | 90.02% | 2.26× |
| ThiNet-30 (Luo et al., 2017) | 68.42% | 88.30% | 3.51× |
| SSS (Huang & Wang, 2018) | 74.18% | 91.91% | 1.45× |
| SFP (He et al., 2018a) | 74.61% | 92.06% | 1.72× |
| CFP (Singh et al., 2018) | 73.4% | 91.4% | 1.98× |
| Autopruner (Luo & Wu, 2018) | 74.76% | 92.15% | 2.05× |
| GDP (Lin et al., 2018) | 71.89% | 90.71% | 2.05× |
| DCP (Zhuang et al., 2018) | 74.95% | 92.32% | 2.26× |
| SSR-L2 (Lin et al., 2019) | 71.47% | 90.19% | 2.26× |
| C-SGD-70 (Ding et al., 2019) | 75.27% | 92.46% | 1.58× |
| C-SGD-50 (Ding et al., 2019) | 74.93% | 92.27% | 1.86× |
| C-SGD-30 (Ding et al., 2019) | 74.54% | 92.09% | 2.26× |
| CNN-FCF-A (Li et al., 2019) | 76.50% | 93.13% | 1.41× |
| CNN-FCF-B (Li et al., 2019) | 75.68% | 92.68% | 1.85× |
| CNN-FCF-C (Li et al., 2019) | 74.55% | 92.18% | 2.33× |
| CNN-FCF-D (Li et al., 2019) | 73.54% | 91.50% | 2.96× |
| Group-HS 1e-5 | 76.43% | 93.07% | 1.89× |
| Group-HS 2e-5 | 75.20% | 92.52% | 3.09× |
| Group-HS 3e-5 | 73.19% | 91.36% | 4.68× |
| Group-HS 4e-5 | 71.08% | 90.21% | 5.48× |

Table 9: Structural pruning result of the ResNet-56 model on CIFAR-10.

| Model | Base acc | Acc gain | #FLOPs reduction |
|---|---|---|---|
| Pruning-A (Li et al., 2016) | 93.04% | +0.06% | 1.12× |
| Pruning-B (Li et al., 2016) | 93.04% | +0.02% | 1.38× |
| Channel pruning (He et al., 2017) | 92.8% | -1.0% | 2.00× |
| NISP-56 (Yu et al., 2018) | N/A | -0.03% | 1.77× |
| SFP (He et al., 2018a) | 93.59% | +0.19% | 1.70× |
| AMC (He et al., 2018b) | 92.8% | -0.9% | 2.00× |
| C-SGD-5/8 (Ding et al., 2019) | 93.39% | +0.23% | 2.55× |
| CNN-FCF-A (Li et al., 2019) | 93.14% | +0.24% | 1.75× |
| CNN-FCF-B (Li et al., 2019) | 93.14% | -1.22% | 3.44× |
| Group-HS 2e-4 | 93.14% | +0.44% | 2.38× |
| Group-HS 2.5e-4 | 93.14% | +0.31% | 3.07× |
| Group-HS 3e-4 | 93.14% | -0.24% | 3.52× |
| Group-HS 5e-4 | 93.14% | -0.91% | 5.63× |

Table 10: Structural pruning result of the ResNet-110 model on CIFAR-10.

| Model | Base acc | Acc gain | #FLOPs reduction |
|---|---|---|---|
| Pruning-A (Li et al., 2016) | 93.53% | -0.02% | 1.19× |
| Pruning-B (Li et al., 2016) | 93.53% | -0.23% | 1.62× |
| NISP-110 (Yu et al., 2018) | N/A | -0.18% | 1.78× |
| SFP (He et al., 2018a) | 93.68% | +0.18% | 1.69× |
| C-SGD-5/8 (Ding et al., 2019) | 94.38% | +0.03% | 2.56× |
| CNN-FCF-A (Li et al., 2019) | 93.58% | +0.09% | 1.76× |
| CNN-FCF-B (Li et al., 2019) | 93.58% | -0.62% | 3.42× |
| Group-HS 7e-5 | 93.62% | +0.44% | 2.30× |
| Group-HS 1e-4 | 93.62% | +0.18% | 3.09× |
| Group-HS 1.5e-4 | 93.62% | -0.08% | 4.38× |
| Group-HS 2e-4 | 93.62% | -0.65% | 5.84× |

