# OpenReview forum: "DeepHoyer: Learning Sparser Neural Network with Differentiable Scale-Invariant Sparsity Measures"
_ICLR.cc/2020/Conference — Accept (Poster)_

### Official Review · AnonReviewer2 · 2019-10-21
**Official Blind Review #2**

**Rating:** 6

**Review:**

The paper is written very nicely and the experiments are convincing (though you can always show more)

In terms of novelty, I shamelessly can say the idea is very simple and the basic form, the Hoyer regularization, was known. That said, I am all in for simple and efficient solutions so I am giving this paper a weak accept for now.

There is not much to ask here (unless I say I want to see how this would work on other backbones and problems). nevertheless, to improve this work, I think the authors need to compare which solution (referring to algorithms in Table1, 2 etc.) is faster/more well-behaved given the combo explained at the bottom of page 5 . This is basically my question/request for the rebuttal.

**Experience Assessment:**

I have published one or two papers in this area.

**Review Assessment: Checking Correctness Of Derivations And Theory:**

I carefully checked the derivations and theory.

**Review Assessment: Checking Correctness Of Experiments:**

I carefully checked the experiments.

**Review Assessment: Thoroughness In Paper Reading:**

I read the paper thoroughly.

---

> ### Author Response · Authors · 2019-11-07
> **Thanks for your positive feedback and addressing your concerns**
>
> Thank you for your positive feedback to our paper. I’d like to emphasize that the main idea of this paper is to find a sparsity-inducing regularizer leveraging the desired property of both the L0 regularizer (scale-invariant, minima along the axis) and the L1 regularizer (almost everywhere differentiable). With these requirements in consideration, we find that Hoyer Square, the square of the traditional Hoyer regularizer, satisfies all the desired properties and behaves as a differentiable approximation to the L0 norm. Extensive experiments are then performed to prove the desired property of Hoyer-Square is truly helpful for both element-wise and structural pruning of DNN models.
>
> The three-stage pruning operations mentioned at the bottom of page 5 is a common practice for DNN pruning. Previous works like iterative pruning (Han et al., 2015b), regularization-based methods (Liu el al., 2015; Wen et al., 2016; Ma et al., 2019), and ADMM (Zhang et al., 2018) etc. all follow similar operations. Since the DeepHoyer regularizer is almost everywhere differentiable, it can be directly added to the original loss function of DNN training and be minimized with SGD (or other gradient-based) optimizers. Thus, the optimization process is as fast as applying L1 regularization, and a lot faster than ADMM which requires complex interplay between multiple objectives. Our experiment results show that DeepHoyer can achieve the lowest sparsity without introducing further complexity in the training process. Among all the pruning methods, SNIP (Lee et al., 2019) is the only one that does not require the 3-stage operations, as it prunes the model at initialization. SNIP might be faster than DeepHoyer, but the final solution it can achieve has much larger amount of parameters (2.5x on MNIST models). As mentioned in the second to last paragraph on page 5, the optimization of DeepHoyer well behaves under SGD. We do not observe any difficulties for training the DNN with DeepHoyer applied. Hope this explanation can address your concern.

---

### Official Review · AnonReviewer1 · 2019-10-21
**Official Blind Review #1**

**Rating:** 6

**Review:**

To enforce sparsity in neural networks, the paper proposes a scale-invariant regularizer (DeepHoyer) inspired by the Hoyer measure. It is simply the ratio between l1 and l2 norm, which is almost everywhere differentiable, and enforces element-wise sparsity. It further proposes the Hoyer measure to quantify sparsity and applies the DeepHoyer in DNN training to train pruned models. The extension of Hoyer-Square is also straightforward.

I generally enjoy simple yet effective ideas. The idea is very straightforward and well intuitive. The paper is well written and easy the follow. The discussion on the Hoyer measure is inspiring and the empirical studies on various different network architecture/datasets compared to several competitive baselines verify the effectiveness of the DeepHoyer model.

Therefore I'm leaning to accept it.

**Experience Assessment:**

I have published in this field for several years.

**Review Assessment: Checking Correctness Of Derivations And Theory:**

I assessed the sensibility of the derivations and theory.

**Review Assessment: Checking Correctness Of Experiments:**

I assessed the sensibility of the experiments.

**Review Assessment: Thoroughness In Paper Reading:**

I read the paper at least twice and used my best judgement in assessing the paper.

---

> ### Author Response · Authors · 2019-11-07
> **Thanks for your positive feedback**
>
> Thank you for your positive feedback to our paper. We are glad that you consider the proposed method effective and find this work inspiring.

---

### Official Review · AnonReviewer3 · 2019-10-29
**Official Blind Review #3**

**Rating:** 6

**Review:**

The paper focuses on sparse neural networks. Typically, l1 regularization is the go-to strategy, however, it is not scale invariant. That is, all weights are affected by the regularization, not only those that are being driven to 0. l0 regularization is theoretically optimal, however, it is not smooth and has no gradients almost everywhere, so it cannot be used for training. As a compromise the paper proposes Hoyer regularization, that is the l1/l2 ratio. The Hoyer regularization has the same minima structure and leads to sparse solutions while being scale invariant, that is it does not affect all weights in the process. Additionally, the paper proposes structured Hoyer regularization. Last, it employs the said regularizations in deep networks: LeNet, AlexNet and ResNet on several datasets: MNIST, CIFAR, ImageNet.

Strengths:
+ The described method is simple, intuitive and straightforward. By applying the said regularization (~ Σ |w_i|/sqrt(Σ w_i^2)), one arrives at seemingly sparser solutions, which is verified in practice.

+ The experiments are extensive and convincing. I particularly like that the authors have used their method with complex and deep models like ResNets, on large scale datasets like ImageNet.

+ The presentation is generally clear and one can understand the paper straightaway.

Weaknesses:
+ The contributions of the paper are rather on the thin side. At the end of the day, Hoyer regularization is taken from another field (compressed sensing) and applied on deep networks. This is also witnessed by some moderate repetition in the writing, e.g., between the introduction and the related work.

+ There are some points where the paper becomes unclear. For instance, in Figure 3 what are the "other methods"?

+ In Figure 1 it is explained that the Hoyer regularization leads to minima along the axis. The gradients then push the models "rotationally". Could this lead to bad multiple local optimal problems? Is there any guarantee that any particular axis will generate better solutions than the other?

All in all, I would recommend for now weak accept. I find the work interesting and solid, although not that exciting.

**Experience Assessment:**

I have read many papers in this area.

**Review Assessment: Checking Correctness Of Derivations And Theory:**

I carefully checked the derivations and theory.

**Review Assessment: Checking Correctness Of Experiments:**

I carefully checked the experiments.

**Review Assessment: Thoroughness In Paper Reading:**

I read the paper at least twice and used my best judgement in assessing the paper.

---

> ### Author Response · Authors · 2019-11-07
> **Thanks for your positive feedback and addressing your concerns**
>
> Thanks for your interest in our paper and your positive and constructive comments. Hopefully this reply can address all your concerns.
>
> For the contribution, the main idea of this paper is to find a sparsity-inducing regularizer leveraging the desired properties of both the L0 regularizer (scale-invariant, minima along the axis) and the L1 regularizer (almost everywhere differentiable). With these requirements in mind, we find that Hoyer Square, the square of the traditional Hoyer regularizer, satisfies all the desired property and behaves as a differentiable approximation to the L0 norm. Extensive experiments are then performed to prove the desired property of Hoyer-Square is truly helpful for both element-wise and structural pruning of DNN models. We believe this paper is exciting as it proves that a simple differentiable approximation to the L0 norm can be used to guide the search of sparse DNNs and outperform much more complex methods that intent to directly optimize with L0 norm, like (Zhang et al., 2018) and (Louizos et al., 2017b).
>
> For Figure 3, we compare our method with many (more than 5 in some figures, and 14 in total) previous works. The main goal of this figure is to show results achieved by DeepHoyer constantly stay above the Pareto frontier of all existing methods rather than some particular methods. We cannot list the names of all the previous methods in the figure due to the limited space. Instead, in the caption of Figure 3, we refer interested readers to Appendix C.3 where all the previous methods and detailed data are listed. Please take a look.
>
> For the local minima problem, in this work we consider the minima along the axis as a desired property of DeepHoyer, because it mimics the minima structure of the L0 norm and can effective lead to sparsity without shrinking the parameter. As mentioned in the second-to-last paragraph on page 5, the recent advance of stochastic gradient descent optimizers provides satisfying performance on large-scale deep learning problem, which itself has a lot of local minima. From the observation in our experiments we believe the tiny bit nonconvexity induced by DeepHoyer does not affect the performance of DNNs. Since there are thousands or millions of parameters in a modern DNN, it’s hard to guarantee which axis will generate better solution. This is why we let DeepHoyer to induce the “rotation” of parameter towards axis but not enforcing a particular one. This will leave the flexibility to the training process to find a solution with optimal sparsity.

---

### Decision · Program_Chairs · 2019-12-19

**Decision:**

Accept (Poster)

**Comment:**

The authors propose a scale-invariant sparsity measure for deep networks. The experiments are extensive and convincing, according to reviewers. I recommend acceptance.